# Tumor Protein D53 (TPD53): Involvement in Malignant Transformation of Low-Malignant Oral Squamous Cell Carcinoma Cells

**DOI:** 10.3390/biomedicines12122725

**Published:** 2024-11-28

**Authors:** Masataka Watanabe, Yoshiki Mukudai, Nodoka Kindaichi, Maki Nara, Konomi Yamada, Yuzo Abe, Asami Houri, Toshikazu Shimane, Tatsuo Shirota

**Affiliations:** Department of Oral and Maxillofacial Surgery, School of Dentistry, Showa University, 2-1-1 Kitasenzoku, Ota-ku, Tokyo 145-8515, Japan; gd21-m029@grad.showa-u.ac.jp (M.W.); tshirota@dent.showa-u.ac.jp (T.S.)

**Keywords:** tumor protein D52 family, tumor protein D53, Akt, squamous cell carcinoma, oral cancer, proliferation, invasion

## Abstract

**Background/Objectives**: The tumor protein D52 (TPD52) family includes TPD52, TPD53, TPD54, and TPD55. The balance between TPD52 and TPD54 expression plays an important role in high-malignant oral squamous cell carcinoma (OSCC) cells. However, the relationship between TPD53 and OSCC cells (particularly low-malignant OSCC cells) remains unclear. In the present study, we investigated the role of TPD53 in the malignant transformation of low-malignant OSCC cells. **Methods**: Temporal changes in the expression of TPD52 family members at the protein and mRNA levels in OSCC cells and normal human epidermal keratinocytes (NHEK) were examined. **Results**: The mRNA expression of *TPD53* increased in HSC-3 and HSC-4 cells in a time-dependent manner. Similar results for protein expression were observed. The effects of TPD53 on anchorage-dependent and anchorage-independent proliferation, cell cycle, invasion and migration, epithelial-mesenchymal transition (EMT), and matrix metalloproteinase (MMP) activities in HSC-3 and HSC-4 cells were assayed. Finally, using the HSC-3-xenograft-nude-mice model, these effects were examined in vivo. Overexpression of *TPD53* increased cell viability and the percentage of cells in the S phase. Furthermore, overexpression of *TPD53* increased cell invasion, migration, and MMP activities, regardless of its effect on EMT. Notably, these effects were more pronounced in HSC-3 than in HSC-4 cells. Overexpression of *TPD53* enhanced tumor formation and growth in mouse xenografts, corroborating the results of in vitro experiments. **Conclusions**: The present study revealed novel and important functions of TPD53 in the proliferation and invasion of low-malignant OSCC cells.

## 1. Introduction

Oral squamous cell carcinoma (OSCC) accounts for approximately 4% of malignant tumors and >90% of oral malignant tumors arising in the mucosal epithelium [1]. The Global Cancer Observatory has estimated 377,713 new cases of OSCC and 177,757 deaths worldwide in 2020 [2]. Although different surgical, radiotherapeutic, and chemotherapeutic approaches have been adopted for treating OSCC, survival rates have remained unchanged over the past 30 years [3,4], and no definitive curing strategy has yet been established. In addition, OSCC is highly malignant, with high morbidity and mortality [5], and involves different genetic backgrounds, such as sex, ethnicity, tumor site, presence of metastases, and viral involvement; numerous tumor types of OSCC cell lines have been recognized with different patterns of biomarkers. Elucidating the mechanisms of cancerous transformation of cells, including that of OSCC cells, is difficult because various genes are tightly linked and polarized in a cascade of migration, proliferation, apoptosis, and metastasis [6].

The tumor protein D52 (TPD52) family of proteins includes TPD52 [7,8], TPD53 [7,9,10,11], TPD54 [10,11], and TPD55 [12]. The first protein identified in this family, TPD52, is overexpressed in breast and lung cancers [13,14]. Except for TPD55, these family members are highly expressed in malignant tumors such as colon [15,16], ovarian [17,18,19], testicular [20,21], and prostate cancers [8,16], as well as lymphoma [22], leukemia [22,23], and brain tumors [15]. Overexpression of *TPD52* in nonmalignant 3T3 fibroblasts induces malignant transformation and increases cell proliferation and anchorage-independent growth [24,25]. Moreover, overexpression of *TPD52* leads to an increase in cell proliferation and phosphorylation of protein kinase B (Akt) in prostate cancer [26,27,28] and protects these cells from apoptosis induced by androgen deprivation [27]. Furthermore, the TPD52 family proteins are components of intracellular nanovesicles, such as intracellular non-membrane bodies responsible for intracellular transport, and affect cell invasion and migration [29]. These reports strongly suggest that members of the TPD52 family may function as novel markers of malignancy.

Recently, we have revealed that TPD52 is highly expressed in both OSCC and surrounding hyperplastic epidermal cells and that TPD54 negatively regulates extracellular matrix-dependent migration and adhesion of OSCC cells [30]. Furthermore, we have reported that the balance between TPD52 and TPD54 expression plays an important role in the proliferation, invasion, metastasis, and survival of high-grade OSCCs under hypoxia [31]. Despite these findings, the roles of TPD52 family proteins in cell proliferation, invasion, and metastasis in low-malignant OSCC cells, such as HSC-3 and HSC-4, remain unclear, and studies on the direct relationship between low-malignant OCSS and TPD53 are scanty.

Therefore, in the present study, we investigated the role of TPD53 in HSC-3, HSC-4, and SAS cells. Additionally, we verified the result using an in vivo mouse xenograft model.

## 2. Materials and Methods

### 2.1. Cell Culture

Human OSCC-derived cell lines, SAS [31,32,33], HSC-3 [32,33,34], and HSC-4 [32,33,34] (human oral squamous cell carcinoma-derived cell lines from tongue cancer, kindly gifted by Dr. Ochiya, Tokyo Medical University), were cultured in high-glucose Dulbecco’s Modified Eagle Medium (HDMEM) with L-glutamine and Phenol red (Wako Chemicals, Osaka, Japan) and supplemented with 10% fetal bovine serum (FBS), 100 U/mL penicillin, and 100 mg/mL streptomycin at 37 °C in an atmosphere of 5% CO_2_ and 100% humidity. Normal human epidermal keratinocytes (NHEKs) were purchased from Promo Cell (Heidelberg, Germany) and grown in an endothelial cell growth medium (Promo Cell), according to the manufacturer’s instructions.

### 2.2. Antibodies

Rabbit monoclonal anti-TPD52 (ab182578) antibody was purchased from Abcam (Branford, CT, USA). Rabbit polyclonal anti-TPD53 (14732-1-AP), anti-TPD54 (11795-1-AP), anti-E-cadherin (20874-1-AP), anti-N-cadherin (22018-1-AP), anti-vimentin (10366-1-AP), and anti-β-actin (20536-1-AP) antibodies were purchased from Proteintech Group Inc. (Rosemont, IL, USA). Rabbit monoclonal anti-Akt1 (#4691), anti-phospho-Akt (S473, #4060), anti-p38 MAPK (#8690), anti-phospho-p38 MAPK (#4511), anti-p44/42 MAPK (Erk1/2) (#4695), anti-phospho-p44/42 MAPK (Erk1/2) (#4370), anti-SAPK/JNK (#9252), and anti-phospho-SAPK/JNK (#4668) antibodies were purchased from Cell Signaling Technology (Danvers, MA, USA). Rabbit polyclonal anti-GFP antibody (598) was purchased from MBL (Aichi, Japan). Rabbit polyclonal anti-HaloTag (G9281) and mouse monoclonal anti-HaloTag (G9211) antibodies were purchased from Promega (Fitchburg, WI, USA). Anti-rabbit IgG, horseradish peroxidase-conjugated whole secondary antibody (NA934V), anti-mouse IgG, and horseradish peroxidase-conjugated whole secondary antibody (NA931) were purchased from Sigma–Aldrich (St. Louis, MO, USA).

### 2.3. RNA Isolation and Real-Time Quantitative Polymerase Chain Reaction (RT–qPCR)

Total RNA was isolated using TRIzol reagent (Life Technologies, Carlsbad, CA, USA), according to the manufacturer’s instructions, and stored at −80 °C until use. RNA (100 ng) was reverse-transcribed into cDNA using a ReverTra Ace qPCR RT Kit (TOYOBO, Osaka, Japan) [32], according to the manufacturer’s instructions. The generated cDNA was subjected to RT–qPCR using a THUNDERBIRD Probe qPCR Mix (TOYOBO), according to the manufacturer’s instructions. The following PCR cycles were used: 95 °C for 1 min, 95 °C for 15 s, and 60 °C for 30 s (49 cycles). TaqMan primers were purchased from Thermo Fisher Scientific (Waltham, MA, USA) (TPD53, catalog no. Hs00914766; TPD52, catalog no. Hs00893105; TPD54, catalog no. Hs00900580; and β-actin, catalog no. Hs00194899). RT–qPCR was performed using a CFX Connect Real-Time PCR Detection System (Bio-Rad, Hercules, CA, USA). Relative gene expression was quantified using the 2^−ΔΔCt^ method and normalized with respect to *β-actin* expression.

### 2.4. Protein Extraction and Western Blotting

Total protein was extracted from cells as previously described [31,33]. For western blotting, 20 µg total protein was separated by sodium dodecyl sulfate–polyacrylamide gel electrophoresis using 4–20% gradient gels (Bio-Rad) and blotted onto polyvinylidene difluoride membranes using iBlot2 (Thermo Fisher Scientific). The membranes were blocked with Tris-buffered saline (Takara Bio, Shiga, Japan) containing 0.2% non-fat dry milk (Cell Signaling Technology) for 1 h. The membranes were then probed with primary antibodies, followed by incubation with secondary antibodies, as previously described [24]. Immunoreactive bands were visualized using an Amersham ECL Prime Western Blotting Detection Reagent (GE Healthcare UK Ltd., Buckinghamshire, UK) on a ChemiDoc XRS Plus Image Lab System (Bio-Rad). Band intensity was quantified using ImageJ 1.54d and normalized with respect to β-actin expression [35]. After detection, the bound antibodies were striped using a Stripping Solution (Wako Chemicals), according to the manufacturer’s instructions. The same blot then was reused for probing using another primary antibody.

### 2.5. Gene Transfection

For knockdown (KD) experiments, small interfering RNA (siRNA) against human *TPD53* (EHU083901) and a negative control siRNA (against firefly luciferase; EHUFLUC) were purchased from Sigma-Aldrich. For overexpression (OE) experiments, the HaloTag-TPD53 expression vector (pFN21AE2169) and HaloTag control vector (G659) were purchased from Promega. The siRNAs and expression vectors were transfected into cells using Lipofectamine 2000 (Thermo Fisher Scientific), according to the manufacturer’s instructions and previous studies [32,33].

### 2.6. MTT Assay

Cells were seeded in 6-well tissue culture plates at a density of 5 × 10^5^ cells/well. After 24 h, cells were transfected with siRNAs or expression vectors. On the day after transfection, cells were seeded in 96-well tissue culture plates at a density of 5 × 10^2^ cells/well. MTT assays were performed after 2 days, as previously described [31,33].

### 2.7. Cell Cycle Assay

Cell seeding and transfection were performed as described for the MTT assay. On the day after transfection, cells were reseeded in 6-well tissue culture plates at a density of 6 × 10^5^ cells/well and grown for 2 days. Cell cycle assay was performed using a commercially available kit (catalog no. A10798, Tali Cell Cycle Kit; Thermo Fisher Scientific), according to the manufacturer’s instructions and previous studies [33,36], on a Tali Image-Based Cytometer (Thermo Fisher Scientific).

### 2.8. Anchorage-Independent Growth Assay

Anchorage-independent growth was assayed using a commercial kit (CytoSelect 96-Well In Vitro Tumor Sensitivity Assay; Cell Biolabs, San Diego, CA, USA), according to the manufacturer’s instructions. A total of 5000 transfected cells were grown on soft agar. After 10 days, cells were examined using a microscope (Eclipse TS100/TS100-F; Nikon, Tokyo, Japan) and photographed with a digital CCD camera (DS-Fil; Nikon), and the colonies were counted (colonies/field). A colony was defined as a cell cluster that was 50 µm in diameter. Cell growth was assayed using MTT assay. All experiments were performed in triplicate.

### 2.9. Cell Invasion Assay

Cell invasion assay was performed using a commercially available kit (CytoSelect 24-Well Cell Invasion Assay; Cell Biolabs, San Diego CA, USA). After transfection, the medium was replaced with fresh medium, and cells were incubated for 24 h. Next, 1.0 × 10^6^ HSC-3 or HSC-4 cells were suspended in serum-free HDMEM. The chamber was incubated with 500 µL fresh serum-free HDMEM supplemented with 10% FBS in a 24-well plate for 48 h. After removing non-invasive cells from the basement membrane, invasive cells were stained and quantified according to the manufacturer’s instructions. Photographs were captured using an Olympus BX51 microscope (Tokyo, Japan) and a microscopic CCD camera (Olympus DP71). Microscopic images were analyzed using a commercial software (Olympus NIS-Elements D3.00, SP6 (Build 539)).

### 2.10. Wound Healing Assay

This assay was performed as previously described [30,31,33,37], with slight modifications. Transfected cells were seeded in a 24-well plate at a density of 2.4 × 10^5^ cells/well and grown for 24 h. Scratches were made using 1-mL pipette tips, and the wells were washed twice with phosphate-buffered saline. Cells were allowed to grow for another 24 h. Then, cells were fixed and stained with crystal violet, and images were captured as described above.

### 2.11. Gelatin Zymography

This assay was performed using a commercially available kit (Gelatin Zymography Kit, Code No. AK47; Cosmo Bio, Tokyo, Japan). After 24 h of incubation post-transfection, the culture medium was replaced by fresh medium. Cells were grown for 48 h, lysed using RIPA buffer (Wako Chemicals), and subjected to gelatin zymographic analysis, according to the manufacturer’s instructions. The densities of degraded bands were measured and analyzed using ImageJ.

### 2.12. Generation of Stable Clones

Halotag-TPD53 was inserted between the SgfI and PmeI sites of pFN28K (Promega), and the construct was transfected into HSC-3 cells. Transfected cells were selected using 1 mg/mL G418 (Sigma-Aldrich), and each single clone was isolated according to the manufacturer’s instructions. To construct a control HaloTag clone, the HaloTag control vector and a large fragment of pFN28K digested using BglII and XbaI, containing a neomycin/kanamycin resistance gene downstream of the SV40 promoter (1/10 of HaloTag vector), were co-transfected into HSC-3 cells. Transfected cells were selected, and single clones were isolated. Halotag-TPD53 expression was analyzed by RT–qPCR. Cells were maintained in HDMEM supplemented with 10% FBS and 1 µg/mL G418. To knockdown *TPD53*, HSC-3 cells were infected with MISSION shRNA lentivirus TurboGFP shRNA for *TPD53* (TRCN0000088393; Sigma-Aldrich) and MISSION control TurboGFP lentivirus (SHC003V; Sigma-Aldrich). Infected cells were selected using 1 mg/mL G418 (for TPD53 shRNA) or puromycin (Sigma-Aldrich; for control TurboGFP lentivirus), and each single clone was isolated according to the manufacturer’s instructions. Expression of TurboGFP and knockdown of *TPD53* were confirmed by fluorescence microscopy and RT–qPCR, respectively. Cells were maintained in HDMEM supplemented with 10% FBS and 1 µg/mL G418 or puromycin.

### 2.13. Mouse Xenograft Model

This study complied with the ARRIVE guidelines and AVMA euthanasia guidelines 2020, was approved by the Institutional Animal Care and Use Committee (approval numbers: 15073 and 224026), and was conducted in accordance with the Showa University Guidelines for Animal Experiments. Four-week-old female BALB/cAJcl-nu/nu mice (average body weight, 20 g) were purchased from Claire Japan (Tokyo, Japan) and maintained under pathogen-free conditions. Each group included five mice (10 mice per experiment for overexpression or knockdown); therefore, 60 mice were used and randomly divided into four groups (i.e., control, TPD53 OE, KD control, and KD TPD53; n = 3 in each experimental group). Approximately 1.0 × 10^6^ cells in 100 µL saline were subcutaneously injected into a unilateral flank, and cancer-bearing mice were maintained for 27 days for tumor growth [32,33]. Movement disorders, anorexia, nausea, and abnormal behavior were expected owing to cancerous growth and metastasis, causing distress and stress in mice. If mice showed loss of body weight by >10% in 7 days, abnormal behavior by excessive stress, such as self-injury, other injury, or damage to the breeding gauge, or if the size of the primary tumor exceeded 4 cm^3^, the experiment was immediately terminated by CO_2_ asphyxiation of mice, even before the set end point (day 27). However, in the present study, no mice exhibited these abnormal behaviors; therefore, all mice were sacrificed after 27 days by CO_2_ asphyxiation. After confirming death by palpation, the tumors were resected, as previously described [32,33]. The day on which cells were injected into mice was set as day 0. Tumor volumes were measured every 3 days. Tumor volume was determined by direct measurement and calculated using the following formula [32,33]:Tumor volume = π/6 × (large diameter) × (small diameter)^2^
(1)

### 2.14. Immunohistochemistry

Resected specimens were fixed with 10% formalin for 24 h, embedded in paraffin, stained with hematoxylin and eosin (Sakura Finetek Japan, Tokyo, Japan), and then immunohistochemically stained for TPD53, HaloTag, and GFP, as previously described [32]. Antigen retrieval was performed with citrate-phosphate buffer (0.01 M, pH 6.0) at 121 °C for 20 min. Endogenous peroxidases were blocked by incubating the samples with 10% H_2_O_2_ (Wako Chemicals) for 15 min. Proteins were blocked using a Protein Block Serum-Free Ready-To-Use (Dako, Carpinteria, CA, USA), according to the manufacturer’s instructions. The sections were incubated at 4 °C overnight with primary antibodies (anti-TPD53 rabbit polyclonal antibody, 1:200 dilution; rabbit polyclonal anti-HaloTag antibody, 1:100 dilution; rabbit polyclonal anti-GFP antibody, 1/200 dilution). The next day, the sections were incubated with secondary antibodies (EnVision+ system-horseradish peroxidase-labeled polymer anti-rabbit; Dako). Finally, sections were reacted with 3,3′-diaminobenzidine + substrate chromogen system (Dako) for color development and examined using a microscope.

### 2.15. Statical Analysis

All experiments were performed at least four times. Statistical analyses were conducted using a Student’s *t*-test and two-way analysis of variance (ANOVA), followed by a post hoc test. Statistical significance was set at *p* < 0.05. ANOVA and post hoc test were performed using KaleidaGraph v.4.5 (Hulinks, Tokyo, Japan), as previously described [32,33,38].

## 3. Results

### 3.1. mRNA Expression of TPD53 Increased in Low-Malignant OSCC Cells in a Time-Dependent Manner

We investigated temporal changes in TPD53 expression in several OSCC cells (SAS, HSC-3, and HSC-4) and NHEK cells (Figure 1 and Appendix A). The mRNA expression of *TPD53* time-dependently increased in HSC-3 and HSC-4 cells, which had low-malignant potentials. In contrast, the mRNA expression of *TPD53* decreased in SAS cells (high-malignant OSCC cells), similar to that observed in NHEK cells. The temporal changes in TPD53 levels were consistent with the results of mRNA expression. These results suggested that TPD53 could play more important roles in low-malignant OSCC cells than in high-malignant OSCC cells and NHEK.

### 3.2. TPD53 Increased Cell Proliferation and Accelerated Cell Cycle Progression in Low-Malignant OSCC Cells

In HSC-3 and HSC-4 cells, overexpression of *TPD53* increased MTT activity (approximately 2-fold), which refers to the number of viable cells, and its knockdown decreased the activity (approximately 0.75-fold) (Figure 2A). Overexpression of *TPD53* increased the percentage of cells in the S phase, indicating accelerated cell cycle progression (Figure 2B,C). However, the effect of *TPD53* knockdown was not significant. These proliferation studies using a monolayer culture suggested an important role of TPD53 in low-malignant OSCC cells. Nonetheless, anchorage-independent proliferation in soft agar (Figure 3) was not significantly altered (*p* > 0.05) by overexpression or knockdown of *TPD53*.

### 3.3. TPD53 Increased Cell Migration and Invasion

The effects of TPD53 on cell migration and invasion were investigated using wound healing and cell invasion assays (Figure 4A–D and Appendix A). Overexpression of *TPD53* increased cell migration (approximately 2-fold), whereas its knockdown decreased cell migration (approximately 0.5-fold). Overexpression of *TPD53* in HSC-3 cells increased cell invasion (approximately 2-fold), and its knockdown decreased invasion (approximately 0.75-fold). Gelatin zymographic analysis (Figure 4E,F and Appendix A) showed that overexpression of *TPD53* increased matrix metalloproteinase (MMP) activities, which were significantly (*p* < 0.05) decreased by knockdown of *TPD53*. Western blot analysis (Figure 4G and Appendix A) indicated that overexpression or knockdown of *TPD53* did not change the expression of E-cadherin, N-cadherin, or vimentin, indicating no involvement in epithelial–mesenchymal transition (EMT). Taken together, TPD53 enhanced cell migration and invasion by activating MMPs but not EMT.

### 3.4. TPD53 Promoted Akt Signaling

We investigated the effect of TPD53 on Akt and three major mitogen-activated protein kinases [MAPKs; p38, extracellular signal-regulated kinase (ERK) and c-Jun N-terminal kinase (JNK)] (Figure 5 and Appendix A). Overexpression of *TPD53* enhanced Akt phosphorylation compared to that in the control. Akt phosphorylation was stimulated by serum; however, knockdown of *TPD53* attenuated this effect. In contrast, overexpression or knockdown of *TPD53* did not affect the phosphorylation of p38, ERK, or JNK after stimulation with serum. These results indicate that the p38, ERK, and JNK pathways have little relationship with the activation of Akt via expression of *TPD53*.

### 3.5. TPD53 Enhanced the Growth of Xenograft HSC-3 in Nude Mice

Based on the results of our in vitro studies, we hypothesized that TPD53 may affect the proliferation, invasion, and migration of low-malignant OSCC cells in in vivo (Figure 6). A timeline of the xenograft experiments is shown in Figure 6A since the tumor sizes of all groups were too small to be measured until day 11. Overexpression or knockdown of *TPD53* had little effect on the body weight of model mice. However, overexpression of *TPD53* significantly increased tumor volumes from day 21 (*p* < 0.05), whereas its knockdown slightly decreased tumor growth (*p* > 0.05). Images of pre- and post-extirpated tumors of day 27 are shown in Figure 6C. Immunohistochemistry revealed stable expression of HaloTag TPD53 and GFP-shTPD53. Notably, TPD53 was relatively highly expressed in the central region of cancer pearl, suggesting that TPD53 may strongly aggravate low-malignant OSCC (Figure 7 and Appendix A).

## 4. Discussion

The physiological and pathological roles of members of the TPD52 protein family in normal and cancerous cells have not yet been completely elucidated. The members of the TPD52 family are expressed in several types of cancers [12,13,14,16,17,18,19,20,21,39,40,41,42], including OSCCs. Overexpression of *TPD52* promotes tumor growth in cancer cells, whereas its knockdown reduces cell migration, invasion [43], and apoptosis [26]. The TPD52 family proteins are components of intracellular nanovesicles, such as intracellular non-membrane bodies responsible for intracellular trafficking, and influence cell invasion and migration [29]. TPD53 regulates the cell cycle and is highly upregulated during the G2–M phase transition [44,45], suggesting a functional association with TPD52 for regulating cell proliferation [45]. However, to the best of our knowledge, the role of TPD53 in OSCCs has not yet been reported. The TPD53 expression increases in a time-dependent manner in HSC-3 and HSC-4 cells [30,31,32,33,34]. In contrast, its expression is low in poorly refabricated and differentiated SAS (highly malignant) [30,31,32,33] and NHEK cells. These results led us to hypothesize that TPD53 plays several important roles in the progression of low-grade OSCCs. In fact, both the mRNA and protein levels of TPD53 increased in a time-dependent manner after reseeding, and the cells began re-proliferation by epigenetic stimulations in the present study. These results indicate an important role of TPD53 in cell proliferation of low-malignant OSCCs.

In the present study, regardless of little effect on anchorage-independent proliferation, overexpression of *TPD53* increased MTT activity and accelerated the cell cycle, whereas its knockdown reduced MTT activity but did not affect the cell cycle. MTT assay directly measures viable cell counts by estimating mitochondrial enzyme activity. In contrast, cell cycle assays determine cell phases by counting the number of chromosomes. Although these assays are commonly used for measuring the number, viability, and/or death of cells, the discrepancy between MTT and cell cycle assays might result from differences in the principles of these assays. The exogenous overexpression of *PrLZ*, a splice variant of *TPD52*, in prostate cancer cells leads to accelerated growth in vitro and tumor formation in vivo [46]. The TPD52 family proteins interact with each other via their coiled-coil domains [10]. The previous reports suggest their mutual interactions that affect cell proliferation and death [7,45].

TPD53 promoted cell invasion and migration, with the effects on HSC-3 cells being more prominent than those on HSC-4 cells. Cell invasion occurs as a result of cell migration [47,48], degradation of extracellular matrix proteins by MMPs [49,50], and increased EMT [47,50,51,52]. We noticed that the overexpression of *TPD53* increased MMP activities; however, it did not significantly affect EMT. These results suggest that TPD53 might play a positive role in regulating the invasion and migration of low-grade OSCCs by enhancing MMP activities but not EMT.

The TPD52 family of proteins enhances Akt phosphorylation [26,27,28]. TPD54 is a negative regulator of Akt signaling [30]. In the present study, Akt phosphorylation was induced by the overexpression of *TPD53*. This highlights the fact that TPD53 and TPD52 are positive regulators of Akt signaling in low-malignant cells [26,27,28], suggesting that TPD53 can enhance the proliferative and invasive potential of cancer cells by increasing MMP activity via the Akt pathway.

In the present study, no significant difference in tumor volume was observed after *TPD53* knockdown in vivo. Therefore, several unknown molecular pathways and/or homomeric and heteromeric interactions via the coiled-coil motifs of TPD52 family proteins may affect malignancy [10,53,54,55,56,57,58].

## 5. Conclusions

The TPD52 family proteins are considered novel targets for cancer therapy as they are expressed in many types of cancer cells, including OSCC cells [59]. We revealed the crucial roles of TPD53 in low-malignant OSCC cells. Therefore, TPD53 can be specifically targeted for therapeutic purposes during the early stage of malignancy. However, further studies are required to investigate the detailed molecular functions of TPD53 and its correlation/association with various cancer markers.

## Figures and Tables

**Figure 1 biomedicines-12-02725-f001:**
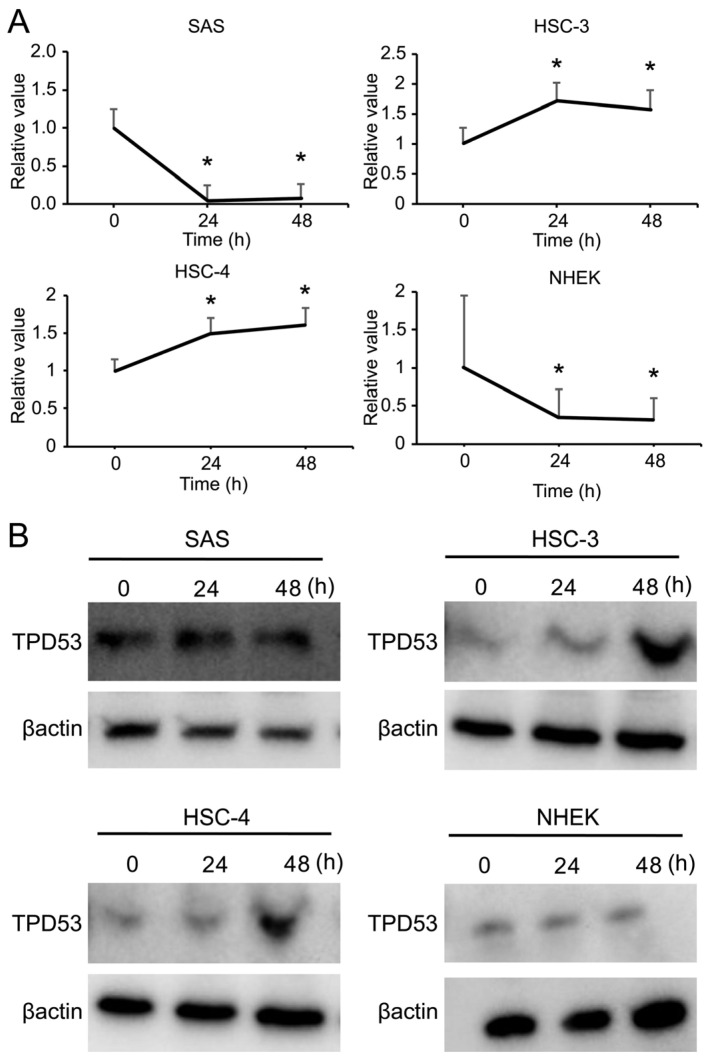
Expression of tumor protein D53 (TPD53) in oral squamous cell carcinoma (OSCC) and normal human epidermal keratinocyte (NHEK) cells. Expression of TPD53 and β-actin at 24 and 48 h measured by real-time quantitative polymerase chain reaction (RT-qPCR) (**A**) and western blotting (**B**). For RT–qPCR, the value at time 0 is designated as “1”, and relative values are shown. * *p* < 0.05 vs. time 0 by Student’s *t*-test. Experiments were repeated thrice, and representative results are shown.

**Figure 2 biomedicines-12-02725-f002:**
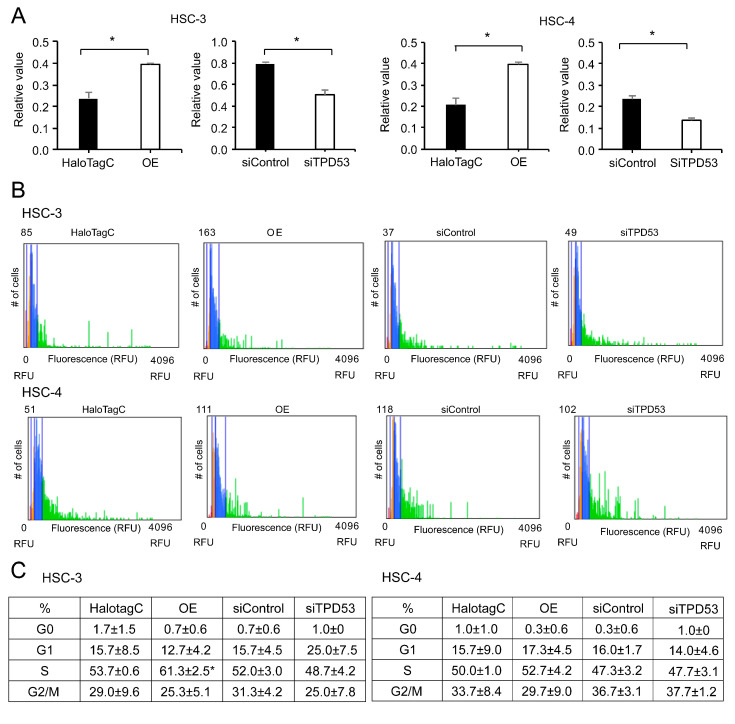
Effects of overexpression or knockdown of *TPD53* on cell proliferation. MTT (**A**) and cell cycle assays (**B**,**C**) for HSC-3 and HSC-4 cells. The histograms indicate the number of cells along the vertical axis and the cell cycle phase along the horizontal axis (red, G0; orange, G1; blue, S; green, G2/M). (**C**). The percentage (±standard deviation) of cells in different phases (G0 and G1, S, and G2/M) for each sample. All experiments were performed thrice, and the representative results are shown. The results were subjected to analysis of variance (ANOVA). * *p* < 0.05 vs. control.

**Figure 3 biomedicines-12-02725-f003:**
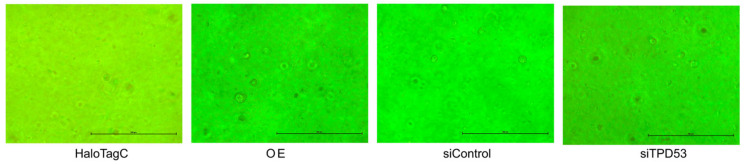
Effects of overexpression or knockdown of *TPD53* on anchorage-independent growth of HSC-3 cells. Cells were transfected with HaloTag-TPD53 (OE) or HaloTag control vector and siRNA for TPD53 (siTPD53) or control. Transfected cells were seeded on soft agar. After 48 h, cells were subjected to colony formation assay. Bar, 200 µm.

**Figure 4 biomedicines-12-02725-f004:**
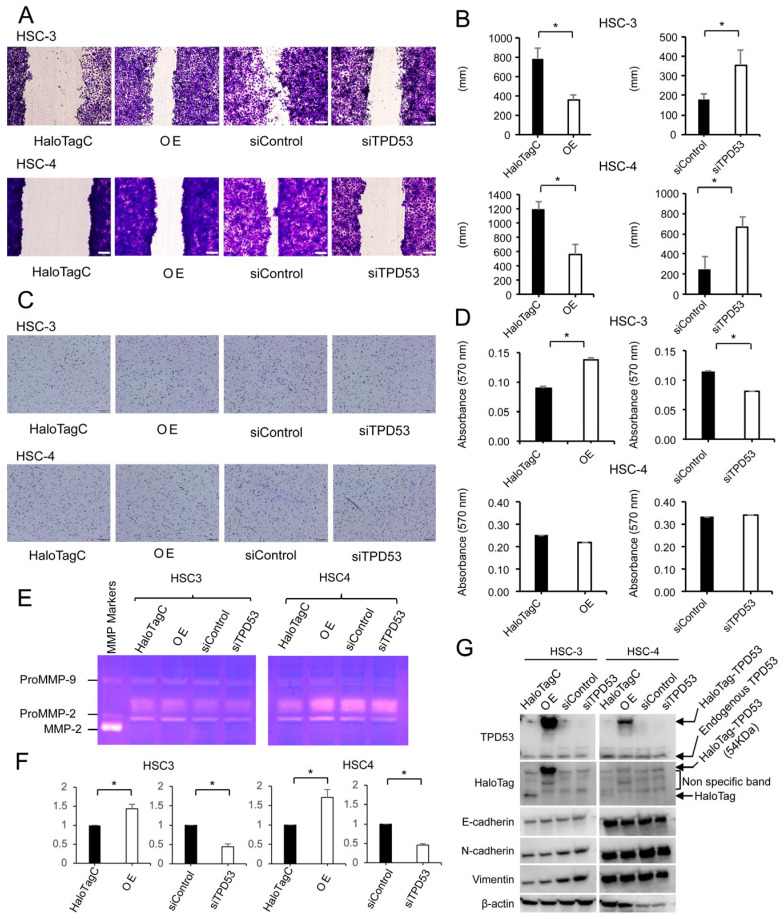
Effects of overexpression or knockdown of *TPD53* on cell migration, invasion, matrix metalloproteinase (MMP) activities, and epithelial–mesenchymal transition (EMT). (**A**,**B**) Wound healing assay. (**C**,**D**) Cell invasion assay. Microscopic images of the invasive cells (**C**), and the absorbance of destained solution from crystal-violet-stained invasive cells (**D**). (**E**) Gelatin zymography for MMP activities. (**F**) Quantification of pro-MMP-2 activity. (**G**) Western blot analysis for assessing EMT (levels of E-cadherin, N-cadherin, and vimentin) and expression of TPD53 and HaloTag. β-actin was used as an internal control. The values in panels (**B**,**D**,**F**) were subjected to ANOVA. * *p* < 0.05 vs. control. Scale bars in panels (**A**,**C**), 200 µm. All experiments were performed four times, and representative results are shown. OE, overexpression.

**Figure 5 biomedicines-12-02725-f005:**
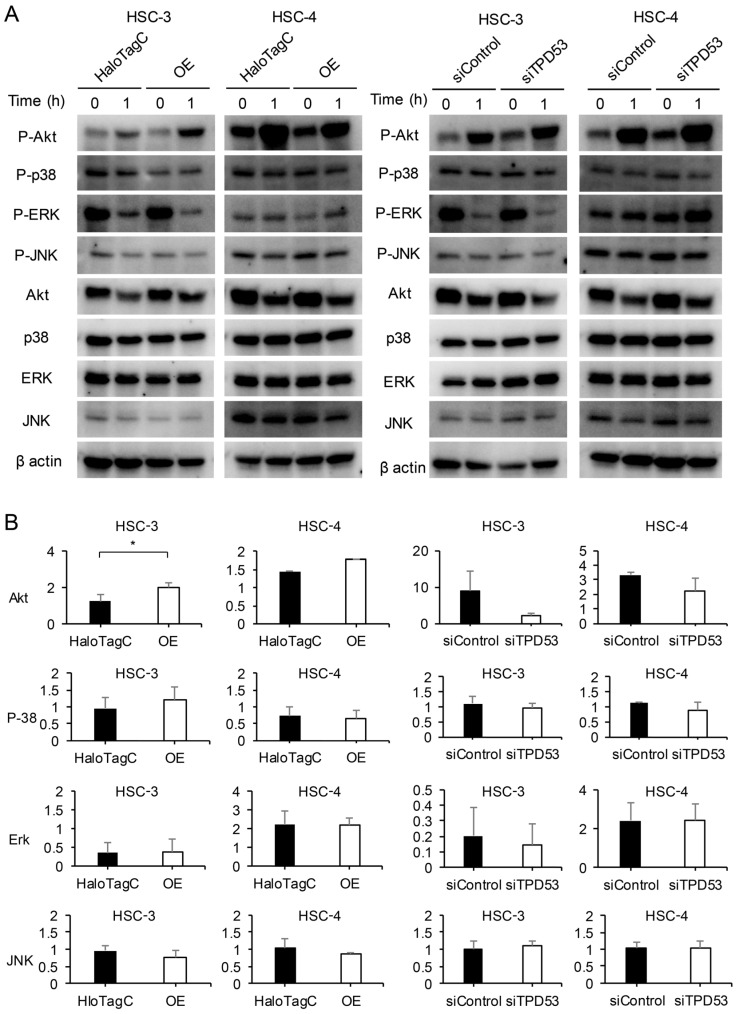
Effect of serum stimulation on the levels of Akt and mitogen-activated protein kinases (MAPKs). Western blot analysis of total and phosphorylated Akt, p38, extracellular signal-regulated kinase (ERK), and c-Jun N-terminal kinase (JNK). Representative images from three replicate experiments are shown (**A**). The ratio of phosphorylation of each MAPK (**B**). * *p* < 0.05 vs. control.

**Figure 6 biomedicines-12-02725-f006:**
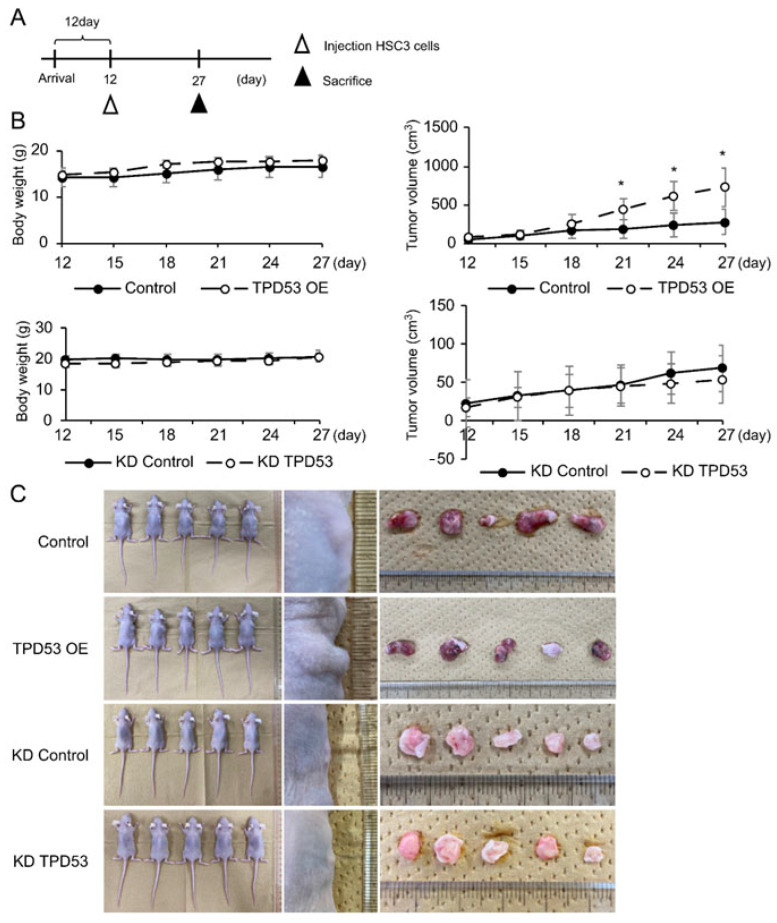
Effects of overexpression or knockdown of *TPD53* on body weight and tumor growth in tumor-xenografted mice. (**A**) Schematic of the tumor-xenograft study. (**B**) The average body weight of mice and average changes in xenograft tumor volumes. * *p* < 0.05 vs. control. (**C**) Images of tumor-xenografted mice and collected tumors on day 27. The experiments were repeated thrice, and representative images are shown.

**Figure 7 biomedicines-12-02725-f007:**
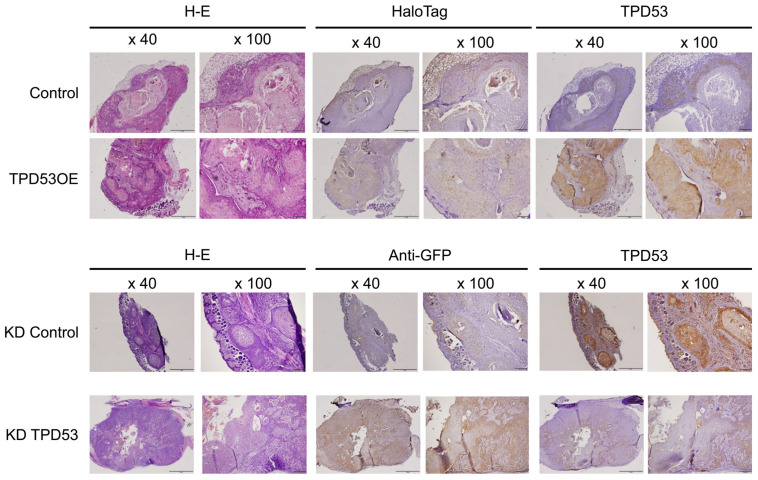
Histopathological images of tumors. Images of the periphery and center of tumors are shown. Bars, 1 mm (40×) and 200 mm (100×). The experiments were repeated thrice, and representative images are shown.

## Data Availability

The data used to support the findings of this study are available from the corresponding author upon request.

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
