# Peer review of "Tumor Protein D53 (TPD53): Involvement in Malignant Transformation of Low-Malignant Oral Squamous Cell Carcinoma Cells"

_biomedicines, 2024, doi:10.3390/biomedicines12122725_

Round 1
Reviewer 1 Report (Previous Reviewer 3)
Comments and Suggestions for Authors
Dear authors,
the manuscript can be published in its present form.
Author Response
We appreciate your helpful and kind suggestions.
Reviewer 2 Report (Previous Reviewer 2)
Comments and Suggestions for Authors
The revised version of the manuscript was significantly improved by the authors. In my opinion, the manuscript can be accepted for publication in the revised form.
Author Response
We appreciate your helpful and kind suggestions.
Reviewer 3 Report (Previous Reviewer 1)
Comments and Suggestions for Authors
See attached.

Round 2
Reviewer 3 Report (Previous Reviewer 1)
Comments and Suggestions for Authors
No further corrections need to be made. However all the western blot data provided in the rebuttal should be included in the supplementary data for transparency.
Author Response
Please see the attachment.

This manuscript is a resubmission of an earlier submission. The following is a list of the peer review reports and author responses from that submission.
Round 1
Reviewer 1 Report
Comments and Suggestions for Authors
Summary: This work describes an exploration of a role for TPD53 expression in malignant transformation in low-malignant oral squamous cell carcinoma cells. Yet, the introduction doesn’t fully support the need for such an investigation, and the experiments, as presented, provide way more questions than answers. Numerous issues detailed below need to be addressed before this article could be considered for publication.
Comments:
Figure 1 looks at TPD53 mRNA and protein expression over time? Time from what? Just being plated? There’s no indication as to what happened at time 0 that would then cause an effect on TPD53 expression.
Also, after these findings, why was the effect of TPD53 on aspects of malignancy even investigated when in the “high-malignant” cell line its expression was decreased/unchanged?
Why was the effect of manipulating TPD53 expression not performed in SAS cells? Why would you think it is important in malignancy at all if its expression was low in these cells?
There is no error (e.g., ± standard deviation) reported in Figure 2C. Was this experiment only performed once? If so, then no conclusions can be drawn. If not, then statistics should have been run to unbiasedly assess the effects on cell cycle.
The soft agar assay is not described in the methods, but results are included in figure S1. The images also look out of focus. From the images presented, it is not clear how any conclusions could be drawn. The authors should also attempt to perform the assay with SAS cells as a positive control to demonstrate the assay is technically working. Also, just out of curiosity, why are those images green? I’ve never seen green soft agar assay images.
The images illustrating the cell migration assay (Figure 3A) results need to have images at time 0 for all control and experimental groups. Otherwise, there is no baseline for which to make comparisons. But assuming the baseline was equivalent for all, how do the authors explain that the siControl group had the highest cell migration rate? Are these cells not essentially the same as the HaloTagC controls?
The text alludes to Figure 3C, but does not definitely list it. It says, HSC-3 cells with overexpressed TPD53 “increased cell invasion”, presumably referencing Figure 3C but that HSC-4 cells did not “form colonies,” which was shown in Figure S1. Figure 3C looks to be showing invasion assays, but they all look identical, despite with the quantification bar graphs show in 3D. A measure of OD 570 here is not sufficient. The authors should actually count the number of invaded cells.
The zymography analysis shown in 3E does not reflect what the text is saying. Moreover, there are no replicates indicated anywhere and no quantification of performed which would even allow a statistical analysis to support those claims.
The authors need to discuss the extra bands for TPD53 and HaloTag in figure 3F. Also, why does their siRNA seemingly only affect HaloTag TPD53 and not endogenous TPD53? The HaloTag shouldn’t even be expressed in the siRNA cell lines. Moreover, why is the TPD53 antibody picking up the HaloTag TPD53 and why is it as such a higher molecular weight than TPD53? Why does the HaloTag specific antibody have two distinct HaloTag bands indicated by two different arrows in those boxes? That antibody should be highly specific. Most importantly, where are the replicate blots? Where is the quantitation? Were all these antibodies assessed on the same blot stripped and re-probed multiple times? If not, then where is the b-actin for each blot?
Many of the same questions need to be addressed for Figure 4A. Where are the replicate blots? Where is the quantitation? Were all these antibodies assessed on the same blot stripped and re-probed multiple times? If not, then where is the b-actin for each blot?
Line 295 states that a statistically significant difference was found at day 21, p > 0.05, but I assume they meant p > 0.05, although the images in Figure 5C certainly do not reflect such a finding. If anything, the TPD53EO excised tumors look much smaller than the control tumors.
Based on the IHC images presented, how do the authors explain the findings that TPD53 expression was highest in the KD control? Shouldn’t both controls have had the same level of TPD53 expression? Also, replicates, quantification, and statistical analyses need to be performed before any conclusions can be drawn from these results.
Small thing, but line 108 says 20 mg protein was used, but very likely the authors meant 20 µg.
Reviewer 2 Report
Comments and Suggestions for Authors
The manuscript by Watanabe and co-workers describes the investigation of the role of tumor protein D52 family member TPD53 in the malignant transformation of low-malignant oral squamous cell carcinoma (OSCC) cells. The research is timely and relevant, taking into account the high incidence and mortality rates of OSCC and the absence of significant improvements in survival rates within the past decades. The manuscript contributes significantly to the understanding of OSCC progression and positions TPD53 as a potential target for early-stage cancer therapy. The manuscript deserves publication.
Specific comments:
Figure 2: It is desirable to enlarge the diagrams in Panel B. It is also necessary to explain different colors in these diagrams in the figure caption.
Figure S1 should be moved from the main manuscript to a separate file containing the supplementary data.
The file attached to the manuscript on submission is named "Raw date of WB" and contains the captions "Raw date of Fig. 1B", ..., etc. Obviously, these captions should be given as "Raw data ..." (not "date"). Can these pictures be regarded as supplementary figures? If yes, they should be numbered, included in the Supplementary Data and cited in the manuscript as usually.
Line 362: The authors mentioned "Supplementary Materials and Methods" here. However, these supplementary materials and methods are not explicitly presented in the submitted files. At the same time, the methodology is properly described in the main manuscript.
I recommend acceptance of the manuscript for publication after minor revision.
Reviewer 3 Report
Comments and Suggestions for Authors
Dear Editor,
the manuscript of Watanabe M. et al. entitled Tumor Protein D53 (TPD53): Involvement in Malignant Transformation of Low-malignant Oral Squamous-cell Carcinoma Cells concerns a very interesting issue about the possibility to explore and find new possible therapeutic targets in TPD52 family proteins. Specifically, this family protein seems to be involved in cell proliferation, invasion and metastasis in low-malignant OSCC; however, their contribution in these processes remains unclear.
Since the topic is very important, solid and well-presented results are expected. On the contrary, the manuscript, in its present form, appears with some critical issues that make the study not sufficient for its publication in Biomedicines. Since the study has the potential to be published, it is recommended to the authors to improve the quality of this manuscript to reconsider the publication of this manuscript in this journal.
Below, only a few critical points that are suggested to be improved by authors:
In the Introduction section: an important and a so method-rich study needs a more in-depth introduction, that also includes the most modern literature.
In the Materials and Methods section: some important information is missing.
- Sometimes authors did not explain in a clear way the methods: there is confusion on how performed experiments on mouse model how the experimental groups have been assembled and how many mice per group. This information is present only in some the figure captions, where the authors mentioned 3 replicates, which justifies the 60 mice. It is recommended to better explain the study design.
- Immunohistochemistry paragraph: it is suggested to improve the sentence: “Resected specimens were fixed with 10% formalin, embedded in paraffin, stained with hematoxylin and eosin (Sakura Finetek Japan, Tokyo, Japan), and then immunohistochemically stained for TPD53, HaloTag, and GFP, as previously described.” As follow: “Resected specimens were fixed with 10% formalin and embedded in paraffin. Morphological analyses were performed thanks to hematoxylin and eosin (Sakura Finetek Japan, Tokyo, Japan) stainings, whereas immunohistochemical stainings for TPD53, HaloTag, and GFP, were carried out as previously described.”. Moreover, authors did not reported time of fixation in formalin: that’s a fundamental point in the histopathological evaluation.
- Authors did not specify from which district the oral squamous cells originate: tongue, gingiva or mucosa?
In Results section: Why authors reported Figure S1 in the main text? “S” doesn’t mean “supplementary”? Moreover, all explanations about results were written unclearly and too concisely to be well understand by readers. This critical issue is also reflected in the discussion section.

No further comments.